

# Testing the effect of bioturbation and species abundance upon discrete-depth individual foraminifera analysis.

Bryan C. Lougheed[1] and Brett Metcalfe[2,3]

1. Department of Earth Sciences, Uppsala University, Sweden.

2. Department of Earth Sciences, Vrije Universiteit Amsterdam, the Netherlands.

3. Laboratory of Systems and Synthetic Biology, Department of Agrotechnology and Food Sciences, Wageningen University & Research, the Netherlands.

*Correspondence to:* B.C. Lougheed (bryan.lougheed@geo.uu.se)

**Abstract.** We use a single foraminifera enabled, holistic hydroclimate-to-sediment transient modelling approach to fundamentally evaluate the efficacy of discrete-depth individual foraminifera analysis (IFA) for reconstructing past sea surface temperature (SST) variability from deep-sea sediment archives, a method that has been used for, amongst other applications, reconstructing El
Niño Southern Oscillation (ENSO). The computer model environment allows us to strictly control for variables such as sea surface temperature (SST), foraminifera species abundance response to SST, as well as depositional processes such as sediment accumulation rate (SAR) and bioturbation depth (BD), and subsequent laboratory processes such as sample size and machine error. Examining a number of best-case scenarios, we find that IFA-derived reconstructions of past SST variability are
sensitive to all of the aforementioned variables. Running 100 ensembles for each scenario, we find that the influence of bioturbation upon IFA-derived SST reconstructions, combined with typical samples sizes employed in the field, produces noisy SST reconstructions with poor correlation to the original SST distribution in the water. This noise is especially apparent for values near the edge of the SST distribution, which is the distribution region of particular interest for, e.g., ENSO. The noise is
further increased in the case of increasing machine error, decreasing SAR and decreasing sample size. We also find poor agreement between ensembles, underscoring the need for replication studies in the field to confirm findings at particular sites and time periods. Furthermore, we show that a species' abundance response to SST could in theory bias IFA-derived SST reconstructions, which can have consequences when comparing IFA-derived SST from markedly different mean climate states. We
provide a number of idealised simulations spanning a number of SAR, sample size, machine error and species abundance scenarios, which can help assist researchers in the field to determine under which conditions they could expect to retrieve significant results.



## 1.0 Introduction

### 1.1 Background

One of the most-studied palaeoclimate signal carrier vessels within deep-sea sediment cores is the carbonate shells of planktonic foraminifera (microscopic, single-celled organisms), which can record the conditions of the ambient water that the foraminifera lived in. These organisms have a lifespan of ~1 month, after which their shells sink to, and are deposited on, the sea-floor. Their short lifespan means that foraminifera microfossil populations retrieved from deep-sea sediment archives can, in principle, reflect past monthly SST dynamics, which is key for reconstructing decadal scale climate

processes, such as El Niño Southern Oscillation (ENSO). However, the technical limits associated with isotope ratio mass spectrometry (IRMS) analysis of foraminifera has traditionally required that many tens of single foraminifera shells to produce a viable measurement, thus averaging out any monthly SST signal. Advances in IRMS have allowed for the analysis of single foraminifera shells

sizes typically found in planktonic populations (Oba and Uomonoto, 1989; Spero and Williams, 1990), which has encouraged researchers to carry out a method commonly referred to as individual foraminifera analysis (IFA) to reconstruct SST variability associated with, e.g., ENSO (Koutavas et al., 2006; Leduc et al., 2009). This method can, in principle, allow for the extraction of a range of monthly SST values from a given interval of a deep-sea sediment archive (i.e. 1 cm discrete depths from a given sediment core).

Using the IFA method, a number of foraminifera are sub-sampled from a discrete-depth's foraminifera population, after which some form of SST proxy method is applied to each foraminifera's carbonate shell to infer individual SST values. Subsequently, an SST distribution can be inferred, and used to indicate past SST variability.

The IFA method depends upon a major assumption, namely that the SST distribution generated from

the sub-sampled foraminifera is a faithful representation of the true distribution of monthly water SST values for a given time interval (i.e. a decadal/centennial/millennial period). However, the ability of discrete-depth IFA to accurately reproduce a time period's true water SST distribution can be clouded by a number of environmental, biological and logistical issues, which can occur in the water domain (pre-deposition), sediment archive domain (post-deposition) and laboratory domain (post-retrieval).

Regarding issues in the water domain, it is possible that a foraminifera species may not continually inhabit a single surface water location or water depth, thus giving a non-continuous record of SST, which can have consequences for, e.g., ENSO reconstructions (Metcalfe et al., 2020). Secondly, a species' foraminiferal abundance through time is not constant and can be influenced by SST itself, which may bias IFA-derived SST distribution reconstructions, which is especially relevant in the case



of ENSO, which itself influences SST. Similarly, long-term absolute shifts in the overall range of SST (e.g. from a glacial to an interglacial world) may cause the water's SST range to shift from one that partially overlaps with a species' preferred temperature range to one that fully overlaps with a species' preferred temperature range. In practical terms, this could lead to an IFA-derived artefactual shift from a relatively narrow apparent SST distribution to a relatively wider apparent SST distribution,

with potential for incorrect interpretation regarding glacial-interglacial SST dynamics.

Issues associated with the sediment archive domain can further cloud IFA-derived SST distributions. Specifically, systematic bioturbation of deep-sea sediment archives means that individual foraminifera with vastly different ages are mixed into single discrete-depth sediment intervals, which is a particular challenge in the current state-of-the-art in IFA, which still relies on the 'average age' of a particular

sediment interval (i.e. it is not yet feasible to decouple single planktonic foraminifera from their discrete depth by systematically dating individual specimens). This practical limitation in turn places an interpretive constraint upon IFA; when foraminifera from vastly different long-term climate states (i.e. multi-millennial) are mixed into the same sediment interval, the IFA-derived SST variability reconstructed from that sediment interval cannot be exclusively assigned to decadal or centennial

changes in inter-annual and intra-annual SST variability (Killingley et al., 1981). For these reasons, it is important to understand the age distribution of foraminifera contained within a discrete-depth sediment interval. For example, it is often assumed that a sediment archive with a sediment accumulation rate (SAR) of, e.g., 5 cm ka$^{-1}$ will have a temporal resolution of $1000/5 = 200$ yr cm$^{-1}$. This assumption is deceptively supported the observation that the mean age of such a sediment

archive increases by ~200 yr every cm. However, downcore increase of discrete-depth mean age is not the same concept as discrete-depth age variance. The distribution of the age contained within a single centimetre of sediment core is governed not only by the SAR, but also by the bioturbation depth (BD), the uppermost depth of the sediment within which bottom-dwelling organisms actively mix the sediment. Following established understanding of bioturbation processes (Berger and Heath,

1968; Pisias, 1983; Schiffelbein, 1984), the $1\sigma$ age value for a single cm of sediment core can be approximated, in the example of a 5 cm ka$^{-1}$ sediment core with a representative BD of 10 cm (Trauth et al., 1997; Boudreau, 1998), as $10/5 \times 1000 = 2000$ yr. In idealised conditions, the corresponding shape of the age distribution for a discrete-depth interval of sediment core will be characterised by an exponential distribution with long tail towards older ages. The average age of the sediment at the top

of the sediment archive will also be similar to the $1\sigma$ age value, as exhibited in $^{14}$C dates of deep-sea core tops which support a BD of between 5 and 10 cm (Trauth et al., 1997; Henderiks et al., 2002), including for the Pacific (Peng et al., 1979; White et al., 2018). It is additionally important to consider the shape of this distribution when comparing IFA-derived SST from an interval of sediment core





(subsampled from a population with a exponential age distribution with a long tail towards older ages) to observational or model SST from specific periods of climate history (i.e. a uniform interval of time).

Finally, issues in the laboratory domain, such as sample size and analytical error, can serve to increase the noisiness of the reconstructed SST distribution and cause interpretive constraints (Killingley et al., 1981; Schiffelbein and Hills, 1984; Thirumalai et al., 2013; Fraass and Lowery, 2017; Dolman and 105 Laepple, 2018; Lougheed, 2020). Consequently, it is important to also consider these processes when considering results derived from discrete-depth IFA analysis.

### 1.2 Experimental Design

Here, we use a computer modelling approach, which uniquely allows all parameters to be known and strictly controlled for, thereby allowing us to create an idealised experimental design with minimised 110 degrees of freedom. Such an approach offers advantages over field-based testing of IFA, where multiple dynamic parameters are unknown, thus leading to increased degrees of freedom and limiting the ability to make interpretative conclusions about the influence of isolated parameters. Our comprehensive modelling approach incorporates quantatitive parameterisations of climate, sediment and laboratory processes. Such a controlled computer model environment allows us to directly 115 compare a known input water SST distribution to a reconstructed SST distribution derived from the corresponding simulated sediment-based IFA. In this way, we can objectively quantify how well discrete-depth IFA functions in a number of strictly controlled, best-case scenarios, allowing its interpretive capacity for the reconstruction of decadal scale SST variability to be evaluated at the most fundamental level.

### 2.0 Method

### 2.1 Approach synopsis and model setup

We carry out a holistic hydroclimate-to-sediment transient modelling approach to test the suitability of discrete-depth IFA for the reconstruction of SST variability. Crucially, our approach includes a quantified representation of both sediment processes (in particular bioturbation) and species 125 abundance, thus building upon previous models and simulation estimations of IFA accuracy where such information was not yet included (Leduc et al., 2009; Thirumalai et al., 2013; Fraass and Lowery, 2017). Our modelling approach is carried out using an offline coupling of two transient models: a single-foraminifera sediment accumulation simulator (SEAMUS; (Lougheed, 2020)) run at a monthly timestep resolution, forced with monthly SST from the TRACE-21ka climate model (He,





2011). We investigate a number of best-case scenarios, concentrating on the time period spanning from 20 ka (BP 1950) up to and including 1989 CE, assuming a hypothetical sediment core location (Fig. 1) at the centre of the Niño 3.4 ENSO region that is used to calculate the Oceanic Niño Index (ONI). While the TRACE-21ka climate model does not necessarily fully capture ENSO processes, we choose this location in the model because of its dynamic SST (Fig. 1), which make it an interesting 135 location to test how inputted monthly SST is reconstructed by the simulated IFA method.

In this study, simulated single foraminifera are incorporated into synthetic sediment archives, the latter of which employ best-case sedimentation conditions whereby representative values for SAR and BD are both kept temporally constant. We assume a best-case scenario where foraminifera perfectly record monthly SST (in this case the TRACE-21ka SST), and we also assume the existence of an 140 ideal proxy method that allows for perfect retrieval of SST data from the single foraminifera. In reality, foraminifera may not continuously record the water temperature at the surface or indeed at the same water depth in general, which further complicates IFA reconstructions of SST dynamics in practice, however, here we seek to test best-case conditions. After carrying out the sediment archive and bioturbation simulation, synthetic single foraminifera are randomly picked from each discrete-145 depth cm interval of simulated core, thereby resulting in virtual IFA. The output of the best-case virtual IFA retrieved from the sediment depth domain can subsequently be directly compared to the inputted SST in the time domain (i.e. TRACE-21ka SST), allowing us to evaluate the current state-of-the-art in IFA at the most fundamental level.

We sum up the sediment model component (SEAMUS) in Section 2.1, and the climate component 150 (TRACE-21ka) in Section 2.2. An overview of our various best-case scenario simulations, as well as their associated run parameters, can be found in Section 2.3.

**2.1 Sediment model component**

We model the sedimentation history of single foraminifera using the the SEAMUS sediment accumulation model (Lougheed, 2020). This stochastic model uses the same established 155 understanding of bioturbation (Berger and Heath, 1968; Pisias, 1983; Bard, 2001) that is also incorporated into previous sediment accumulation models (Trauth, 1998, 2013; Dolman and Laepple, 2018), but differs in model execution in that it is explicitly designed for the purpose of modelling single foraminifera, thus making it a suitable sediment model for use in this IFA evaluation study. The stochastic nature of the model is ideal for simulating bioturbation of single foraminifera, which is in 160 itself a stochastic process. Furthermore, this bioturbation model is capable of receiving temporally dynamic input for all parameters. Our period of interest spans 20 ka BP to 1989 CE, so we have run



the SEAMUS model from 30 ka BP to 1989 CE to provide sufficient model spin-up for our period of interest. The model is run using a monthly timestep resolution, whereby single synthetic foraminifera are generated at each time-step and added to the top of the sediment archive after which the BD of the sediment archive is uniformly mixed. All simulations are run with an appropriate BD of 10 cm, following previous studies (Trauth et al., 1997; Boudreau, 1998). Some of our model run scenarios assume a temporally constant foraminiferal abundance, in such cases we assign a constant per timestep foraminiferal abundance that results in $10^4$ foraminifera per cm of sediment (i.e. the prescribed per timestep abundance is higher in the case of higher SAR and vice-versa). In the case of model runs with temporally dynamic foraminiferal abundance, the amount of foraminfiera per cm that will result in 100 foraminifera per timestep (i.e. month) for the given SAR is simulated, allowing temporal (i.e. monthly) changes in abundance to be modelled with sufficient statistical power (i.e. if relative abundance of the species drops from 0.56 to 0.55 then it will result in one less foraminifera of the species being simulated for a timestep). All of our model run scenarios are carried out using 100 ensemble runs in SEAMUS, thus fully capturing (for 100 percentiles) the stochastic nature of bioturbation (i.e. the fact that no two sediment archives formed under the same conditions will be exactly alike). Subsequently, four separate randomised 'picking' scenarios are carried out on each of the 100 ensembles, whereby 50, 100, 500 or $10^4$ synthetic foraminifera are randomly picked from each discrete 1 cm depth slice of the synthetic core, whereby the picker is assumed to have perfectly identified the species in all cases, thus avoiding challenges associated with species mis-identification (Pracht et al., 2019). Finally, in some scenarios we add Gaussian noise of ±1°C to the SST of all simulated foraminifera, to mimic proxy uncertainty. All ensemble runs were performed using a Linux computer cluster provided by the Swedish National Infrastructure for Computing (SNIC) at the Uppsala Multidisciplinary Centre for Advanced Computational Science (UPPMAX).

## 2.2 Climate model component

Monthly SST forcing for the SEAMUS model is sourced from the TRACE-21ka transient climate simulation (He, 2011), specifically using the surface temperature data for the TRACE-21ka grid cell centred on the coordinates 1.86° N and 146.25° W. This grid cell, at the centre of the Niño 3.4 ENSO region used for calculating the ONI-index, is ideal for our synthetic core simulation as it is characterised by large variation in the model's inter-annual seasonal surface temperature (Fig. 1a), somewhat analogous to, e.g., ENSO. Furthermore, the grid cell also captures the glacial-interglacial SST transition (Fig. 1b), as well as typical TRACE-21ka transient changes in ENSO-like SST variability, as shown by the 1.5-7 yr filtered 100 and 1000 year moving 1σ of SST (Fig. 1c). This filtering approach has previously been used to identify ENSO-like variability in TRACE-21ka for the



Niño 3.4 region (Liu et al., 2014). While the model variability is itself of course not a true replication of the real ENSO signal, it nonetheless offers an interesting analogous timeseries of inter-annual changes in SST variability with which to test the efficacy of the IFA method in reproducing said SST variability.

The TRACE-21ka dataset is the result of a fully-coupled Community Climate System Model
(CCSM3) simulation with T31_gx3 grid resolution that uses transient forcing changes in both greenhouses gases, orbital driven insolation variations, ice sheet evolution (ICE-5G) and associated meltwater fluxes for a non-accelerated atmosphere-ocean-sea ice-land surface coupling. The TRACE-21ka dataset begins at 22 ka, whereas our SEAMUS run starts at 30 ka. The reason for this difference is that we provide an extra 10 ka of spin-up time for the SEAMUS model, which is important in cases
of very low SAR (e.g. $\leq 5$ cm ka$^{-1}$). In order to provide SST data for synthetic foraminifera generated between 30 ka and 22 ka, the oldest 1500 years contained within the TRACE-21ka dataset are repeated from 22 ka to 30 ka. Such an approach obviously does not represent an accurate picture of the climate between 30 ka and 22 ka, but it has no practical consequences for the particular purpose of our study, which is to compare a given climate input signal in the time domain to the subsequent
signal recorded by single foraminifera in the sediment depth domain. Furthermore, our period of study interest spans the past 20 ka.

### 2.3 Model run settings

We carry out a number of best-case scenarios, with each scenario being subject to 100 ensemble runs to capture the full stochastic range resulting from the sedimentation, bioturbation and picking
processes. We run SAR scenarios for 5, 10 and 40 cm ka$^{-1}$. In the figures in the main text, we concentrate on the 10 cm ka$^{-1}$ scenarios only. The corresponding figures for the 40 cm ka$^{-1}$ and 5 cm ka$^{-1}$ scenarios, the latter of which may be more realistic for much of the Pacific, can be found in the supplement. Each of the three SAR scenarios is first subjected to 100 ensemble runs with constant foraminifera abundance and a perfect SST proxy, a second set of 100 ensemble runs is then carried
out with constant abundance and added $\pm1°C$ Gaussian noise on the SST proxy, a third set of 100 ensemble runs is carried out with dynamic abundance and a perfect SST proxy, and a final set of 100 ensemble runs is carried out with dynamic abundance and $\pm1°C$ Gaussian noise on the SST proxy. All of the aforementioned 1200 ensembles are each subjected to randomised picking for 50, 100, 500 and $10^4$ foraminifera per cm of sediment core depth.

As described in the previous paragraph, some of our scenarios incorporate dynamic foraminiferal abundance in order to investigate the effect of changes in species abundance upon IFA-derived





reconstructions. In these scenarios, we use a hypothetical transfer function (Fig. 2a) to assign a per timestep abundance to our simulated foraminifera species. This theoretical transfer function is purely demonstrational, and is used to gain insight into how a given abundance response influences IFA reconstructions of SST variability. Timestep abundance is calculated as a by applying the function to the corresponding TRACE-21ka SST for the timestep. This approach allows us to quantify how a known species abundance response to SST could systematically bias an IFA-derived SST distribution. Consider, for example, a theoretical time interval whereby the true monthly SST data are normally distributed, as in the theoretical example in Fig. 2b. In such a case, an IFA-derived SST distribution using a species characterised by our SST transfer function would be biased towards warmer temperatures and, furthermore, the shape of the IFA-derived SST distribution would be skewed, as shown in the abundance-modified profile in Fig. 2b.

### 3.0 Results & Discussion

### 3.1 Downcore, discrete-depth IFA standard deviation

Numerous studies have concentrated on subsampling numerous individual foraminifera from the same discrete-depth interval of a sediment core, from which the 1σ value of the SST (or a proxy equivalent thereof) of those foraminifera is calculated to infer SST variability for a particular time period, whereby a greater 1σ value is assumed to indicate increased SST variability due to, e.g., ENSO (Koutavas et al., 2006; Koutavas and Joanides, 2012; Rustic et al., 2015). To evaluate such an approach, we compare the 1.5-7 yr filtered 1000 year moving 1σ of SST in the time domain (Fig. 1c) to ensembles of SEAMUS runs carried out under various sediment and picking conditions within a 10 cm ka$^{-1}$ scenario (Fig. 3 and Fig. 4). The equivalent figures for the 40 cm ka$^{-1}$ and 5 cm ka$^{-1}$ scenarios, the latter of which may be more representative for the open ocean areas of the Pacific (Olson et al., 2016; Metcalfe et al., 2020), can be found in the supplement.

We find that the discrete-depth, downcore 1σ value reconstructed using IFA analysis for the simulated 10 cm/ka scenarios varies greatly between all of the 100 ensemble runs in the case of IFA sample sizes typically used in the field, i.e. between 50 foraminifera (Fig. 3a-b; Fig. 4a-b) and 100 foraminifera (Fig. 3c-d; Fig. 4c-d) individual foraminifera being picked per cm. This poor reproducibility between ensemble runs is a result of noise generated by small sample sizes in combination with systematic bioturbation. The practical consequence of this poor reproducibility is that, in the case of typical sample sizes used in the field (50-100 foraminifera), none or very few of the 100 ensemble runs result a significant correlation (defined here as $r^2 \geq 0.6$ and $p \leq 0.05$) between the IFA-derived downcore 1σ SST signal and the 1.5-7 yr filtered TRACE-21ka 1000 year moving 1σ



(Table 2), for the period 18 ka to 12 ka, a period of dynamic ENSO-like variation in the TRACE-21ka
SST. Furthermore, the wide 95.4% band of ensemble downcore 1σ SST values demonstrates a
practical challenge for studies that compare decadal and centennial SST variability from two distinct
time periods by comparing, e.g., a late glacial sediment slice's 1σ SST value to a late Holocene
sediment slice's 1σ SST value. In such cases, our model results suggest that, for the aforementioned
typical sample sizes deployed in the field (50-100 foraminifera), random chance may lead to any
number of possible apparent outcomes regarding the relative apparent SST variablity of the late
glacial and the late Holocene.

We do find, however, that greatly increased sample size, higer SAR and reduced measurement error
can all significantly improve the probability of a given ensemble's IFA-derived downcore 1σ SST
exhibiting significant correlation with the TRACE-21ka SST variation (Table 2). We must stress,
however, that our best-case scenarios involve constant SAR and BD, whereas real world conditions in
the field are inherently dynamic and would, therefore, be more challenging. Additionally, we note that
the improved correlation in the case of larger samples size does not correspond to a good reproduction
of the *absolute* values of the SST variation as indicated by the TRACE-21ka SST ENSO-type
variation. Even in the case of an extreme best-case scenario where it is possible to find, pick and
analyse $10^4$ foraminifera per cm, the absolute values of the ENSO-type variation derived from IFA are
systematically greater than that of the TRACE-21ka SST ENSO-type variation (Fig. 3g and Fig. 4g),
despite good correlation (Table 2). This offset in absolute values can be due to the fact that the 1.5-7
yr filtered, 1000 year smoothed TRACE-21ka standard deviation is reflecting a different integration
of the time than the 1σ data retrieved from discrete-depth IFA. The former is based on a smooth of
uniform time, whereas the latter is represents a population of foraminifera with a long-tailed age
distribution. The absolute offset between the two signals is further increased in the case of machine
error on the IFA SST analysis (Fig. 3h and Fig. 4h), thus highlighting the importance of accurately
quantifying uncertainties in the analytical process.

### 3.2 Discrete-depth IFA distribution analysis

Many IFA studies have gone beyond studying a discrete depth's 1σ SST value and have branched into
more forensic studies of a discrete depth's IFA-derived SST distribution. These studies have focussed
on analysing the shape of said distribution using various statistical tools, including skewness analysis
of histograms (Leduc et al., 2009; Khider et al., 2011), as well as quantile-quantile (Q-Q) plots (Ford
et al., 2015; White et al., 2018; Thirumalai et al., 2019; Rongstad et al., 2020; White and Ravelo,
2020). Such analysis can reveal apparent shifts in the shape of the downcore, IFA-derived SST



distribution, which the aforementioned studies have attributed to changes SST changes in the water caused by ENSO-type climate variability.

Here, we compare the monthly TRACE-21ka SST data for the 18 ka to 17 ka period to our 100 ensembles of simulated IFA SST for our 10 cm ka⁻¹ scenario, taking in each ensemble the 1 cm
discrete-depth with a median age closest to 17.5 ka. We show 100 ensembles with no analytical error and constant abundance (Fig. 5), 100 ensembles with ±1° C analytical error and constant abundance (Fig. 6), 100 ensembles with ±1° C analytical error and dynamic abundance (Fig. 7), and 100 ensembles with ±1° C analytical error and dynamic abundance (Fig. 8). In all cases in our 10 cm ka⁻¹ scenario, we find that sample sizes typically associated with IFA in the field (50-100 foraminifera)
produce high levels of noise, leading to low reproducibility from one ensemble to the next (panels a and d in Figs. 5-8). As expected, the 5 and 40 cm ka⁻¹ scenarios (see supplemental figures) result in lower and higher reproducibility, respectably. In practical terms, these results suggest that if one were to, at the same coring location, retrieve multiple sediment cores and carry out discrete-depth IFA, it is possible that different outcomes would be produced each time, each with sub-optimal correspondence
to the true SST distribution in the water. Furthermore, as the level of noise increases with lower SAR, one has to be additionally careful when comparing IFA results from sites with markedly different SAR.

We also find that the IFA method has a tendency for noisy over- or undersampling of the tails of the true SST distribution in the case of typical sample sizes (50-100 specimens) used in the field (panels b
and e in Fig. 5-8). This effect can be attributed to the fact that there is a low occurrence of individual foraminifera within the population that record more extreme SST, and small sample sizes are likely to either miss such foraminifera altogether (i.e., -100% oversampling), or, in the case of a single such foraminifera being picked within the sample, significantly over-represent extreme SST within the sample (in some cases >500% oversampling). This effect has practical consequences for
interpretations made within IFA studies, seeing as the tails of the SST distribution are the region of interest when reconstructing the presence of, e.g., extreme ENSO events (Koutavas et al., 2006; Rustic et al., 2015). This noisy under- or oversampling of the distribution tails by IFA also translates directly to sample Q-Q plots (panels c and f in Fig. 5-8), which are commonly used in IFA studies to investigate the population distribution (Ford et al., 2015; Rongstad et al., 2020). This level of noise in
the tails increases substantially in the case of increased analytical error, i.e. when one compares panels a-f in Fig 5 (without simulated analysis error) and Fig. 6 (with ±1°C simulated analysis error). We furthermore find that even larger sample sizes involving 500 foraminifera are also prone to noisy under- or oversampling in the tails, especially in the case of analytical error (panels g, h, and i in Fig.



6). We also note that the tendency for under- and oversampling in the tails is greatly increased in the case of lower SAR and somewhat reduced in the case of higher SAR (see supplemental figures for 5 cm ka$^{-1}$ and 40 cm ka$^{-1}$ SAR scenarios). Even in the case of sample sizes of $10^4$ foraminifera in our 10 cm ka$^{-1}$ scenario (panels j, k and l in Figs. 5 and 6) we also find sub-optimal agreement with the TRACE-21ka SST distribution in the tails. This disagreement is not due to noise, but due to the fact that we emulate the current state of the art, whereby SST from a uniform interval of time (in our case 18 ka to 17 ka) is compared to a sample of foraminifera retrieved from sediment with a population characterised not by a uniform distribution of time, but an exponential distribution with a long tail towards older ages.

Finally, we investigate the influence of temperature-induced species abundance changes upon IFA-derived SST distributions. Our 10 cm ka$^{-1}$ simulations that have been run using the temperature abundance transfer function in Fig. 2a are shown in Fig. 7 (without analytical noise) and Fig. 8 (with analytical noise). We find that in all cases, the IFA-derived SST distribution is biased towards too warm values when compared to the TRACE-21ka SST distribution (panels a, d, g and j in Fig. 7 and Fig. 8). This bias can also be visualised as an oversampling of warmer values (panels b, e, h, k in Fig. 7 and Fig. 8), or bias in a Q-Q plot (panels c, f, i, l in Fig. 7 and Fig. 8). We demonstrate that a species' abundance response to temperature can inherently bias IFA-derived reconstructions of SST distribution, which could have practical consequences for studies in the field. For example, the results in studies that compare IFA-derived SST distributions from significantly differing mean climate states (White et al., 2018; White and Ravelo, 2020), may be (partially) attributable to a species' temperature abundance response to the dominating SST profile associated with the differing climate states. Our results demonstrate the importance of incorporating understanding of past temporal changes of species abundance and its relationship to past SST.

### 4.0 Conclusion & Outlook

Our best-case modelling study reveals a number of challenges which inhibit the efficacy of discrete-depth IFA in producing reconstructions of past SST distribution, the latter of which is paramount in reconstructing, e.g., past ENSO-type climate dynamics. Firstly, we find that bioturbation of sediment archives, combined with typical sample sizes employed in IFA-based studies, can lead to noisy IFA-derived SST distribution reconstructions. This noise leads to poor reproducibility with a potential for artefactual results. We would like to reiterate that our best-case model scenarios are possibly not representative for field studies that have been carried out, and it is entirely possible that existing studies have been retrieved from areas with a BD that is significantly more or less than the global average of 10 cm. Consequently, our model results may either over- or understate challenges relevant



to IFA. We propose, therefore, that studies in the field can improve quantification of the total error on IFA- reconstructions using three main approaches: (1) Quantification of real-world sedimentological parameters (SAR, BD) and foraminiferal parameters (abundance, temperature sensitivity) at the core site. (2) Ensemble-based forward model studies, as detailed in this study using best-case scenarios, can be run using the sediment and foraminiferal parameters present at the core site. This approach will help estimate the total stochastic error associated with the IFA-derived reconstruction. Care must be taken to include uncertainties regarding time-domain estimations of SAR, BD, species abundance, and analytical uncertainty. (3) Replication studies in the field (essentially a real-world ensemble approach) to help to further understand of the the stochastic noise involved with IFA reconstructions.

We furthermore have shown in our best-case study that a species' abundance response to SST can inherently bias IFA-derived reconstructions of past SST variability. We propose that the coupling of a single foraminifera sediment model approach to foraminiferal ecological models (Lombard et al., 2011; Roche et al., 2018; Metcalfe et al., 2020) could further help to constrain the total uncertainty associated with IFA-derived SST reconstructions.

We have also demonstrated that observed or model SST from uniform periods of time (as humans are accustomed to using) cannot directly be compared to IFA-derived SST which is retrieved from a population with an age distribution characterised by an exponential distribution with a long tail towards older ages. Subsequently, we propose that researchers adjust observational or model SST data to integrate an exponential representation of time when comparing to IFA-derived SST.

## Author contributions

BCL and BM conceived the study. BCL executed the model runs and wrote the manuscript, with input from BM.

## Acknowledgements

BCL acknowledges Swedish Research Council (Vetenskapsrådet – VR) Starting Grant number 2018-04992. The Swedish National Infrastructure for Computing (SNIC) at the Uppsala Multidisciplinary Centre for Advanced Computational Science (UPPMAX) provided computing resources for parallel ensemble runs. Jesper Sjolte and Feng He are thanked for help in locating the correct TRACE-21ka model run file.





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



**Table 1.** Overview of SAR and number of picked specimens in select IFA studies (including non-ENSO studies). Region codes are as follows: WEP – Western Equatorial Pacific; CEP – Central Equatorial Pacific; EEP – Eastern Equatorial Pacific; EEI – Eastern Equatorial Indian Ocean; SIO – Southern Indian Ocean; ARA – Arabian Sea. We have estimated the 1σ value of age in 1 cm of sediment based on the SAR and a BD of 10 cm (Boudreau, 1998), using the following calculation based on (Berger and Heath, 1968): BD/SAR×1000, where SAR is entered in cm ka$^{-1}$ and BD in cm.

| Core(s) | Study | Region | Approximate SAR (cm ka$^{-1}$) | Estimated 1σ value of age in 1 cm (yr) | Specimens picked per discrete interval (#) |
|---|---|---|---|---|---|
| MGL1208-14MC and 12GC | (White et al., 2018) | CEP | ~2.5 | 4000 | 70 ~ 90 |
| ODP 806 | (Ford et al., 2015) | WEP | ~ 3 | 3300 | 60 ~ 70 |
| ODP 849 | (Ford et al., 2015) (White and Ravelo, 2020) | EEP | ~ 4 | 2500 | 60 ~ 70 |
| KNR195-5 MC42 | (Rustic et al., 2015) | EEP | ~12 | 830 | 55 |
| MD02-2529 | (Leduc et al., 2009) | EEP | ~40 | 250 | 65 ~ 90 |
| V21-30 | (Koutavas et al., 2006) (Koutavas and Joanides, 2012) | EEP | ~12 | 830 | 50 |
| MD98-2177 | (Khider et al., 2011) | WEP | ~70 | 150 | 60 ~ 90 |
| SO189-119KL | (Thirumalai et al., 2019) | EEI | ~20 | 500 | 55 ~ 65 |
| SO189-39KL | (Thirumalai et al., 2019) | EEI | ~ 37 | 270 | 55 ~ 65 |
| GeoB 10038-4 | (Thirumalai et al., 2019) | EEI | ~9 | 1100 | 55 ~ 65 |
| GeoB 10053-7 | (Thirumalai et al., 2019) | EEI | ~35 | 290 | 55 ~ 65 |





| NIOP 905P | (Ganssen et al., 2011) | ARA | ~20 | 500 | 30 ~ 40 |
|---|---|---|---|---|---|
| 64PE-174P13 | (Scussolini et al., 2013) | SIO | ~ 1.2 | 8330 | 20 ~ 30 |






**Table 2.** Statistical testing of the ability of the downcore sediment 1σ record to reflect millennial-scale temporal trends in palaeo-ENSO. Shown in the table, for each scenario, is the number of the 100 ensemble runs whereby the Pearson correlation coefficient between the downcore sediment 1σ record and the 1.5-7 yr filtered, 1000 year smoothed TRACE-21ka standard deviation exhibits an $r^2 \geq 0.6$ and $p \leq 0.05$. Correlations are carried out for the 18 ka to 12 ka period, a period of dynamic signal for the 1.5-7 yr filtered, 1000-year smoothed TRACE-21ka standard deviation.

| # forams picked | Constant foraminifera abundance | | | | | | Dynamic foraminifera abundance | | | | | |
|---|---|---|---|---|---|---|---|---|---|---|---|---|
| | 5 cm ka⁻¹ BD 10 cm no error | 5 cm ka⁻¹ BD 10 cm ±1°C err. | 10 cm ka⁻¹ BD 10 cm no error | 10 cm ka⁻¹ BD 10 cm ±1°C error | 40 cm ka⁻¹ BD 10 cm no error | 40 cm ka⁻¹ BD 10 cm ±1°C error | 5 cm ka⁻¹ BD 10 cm no error | 5 cm ka⁻¹ BD 10 cm ±1°C error | 10 cm ka⁻¹ BD 10 cm no error | 10 cm ka⁻¹ BD 10 cm ±1°C error | 40 cm ka⁻¹ BD 10 cm no error | 40 cm ka⁻¹ BD 10 cm ±1°C error |
| **50** | 0 | 0 | 0 | 0 | 0 | 0 | 0 | 0 | 4 | 0 | 28 | 0 |
| **100** | 0 | 0 | 6 | 0 | 100 | 5 | 1 | 1 | 49 | 3 | 100 | 55 |
| **500** | 6 | 2 | 100 | 93 | 100 | 100 | 60 | 21 | 100 | 100 | 100 | 100 |
| **10⁴** | 100 | 99 | 100 | 100 | 100 | 100 | 100 | 100 | 100 | 100 | 100 | 100 |




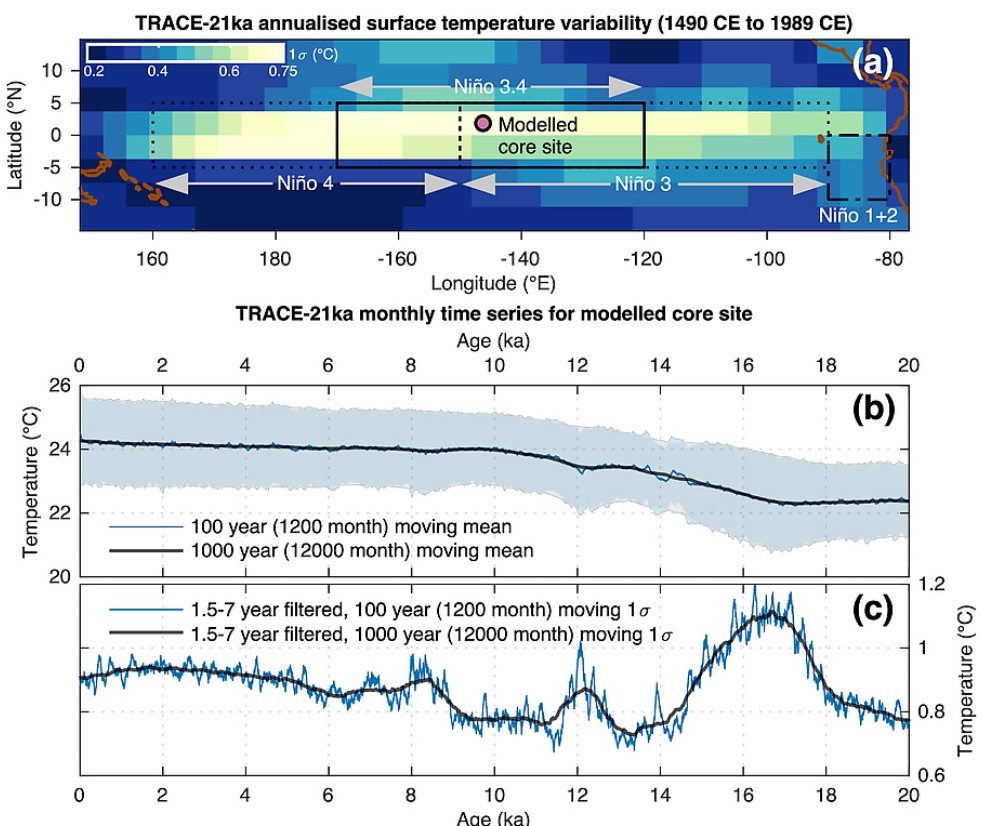

**Figure 1.** Overview of the modelled core site location and associated TRACE-21ka data. Panel a: The lcation of the modelled sediment core site superimposed upon the standard deviation of annualised SST from the TRACE-21ka for the 500 year period between 1490 CE and 1989 CE. Also shown for reference are the Niño regions 1+2, 3, 3.4 and 4. Panel b: 100 year (1200 month) and 1000 year (12000 month) moving mean the monthly TRACE-21ka SST data for the modelled sediment core site. Also shown in light blue and light grey are the moving ±1σ envelopes respectively associated with the moving 100 year (1200 month) and 1000 year (12000 month) windows. Panel c: 100 year (1200 month) and 1000 year (12000 month) moving 1σ of the 1.5-7 year filtered monthly SST data.


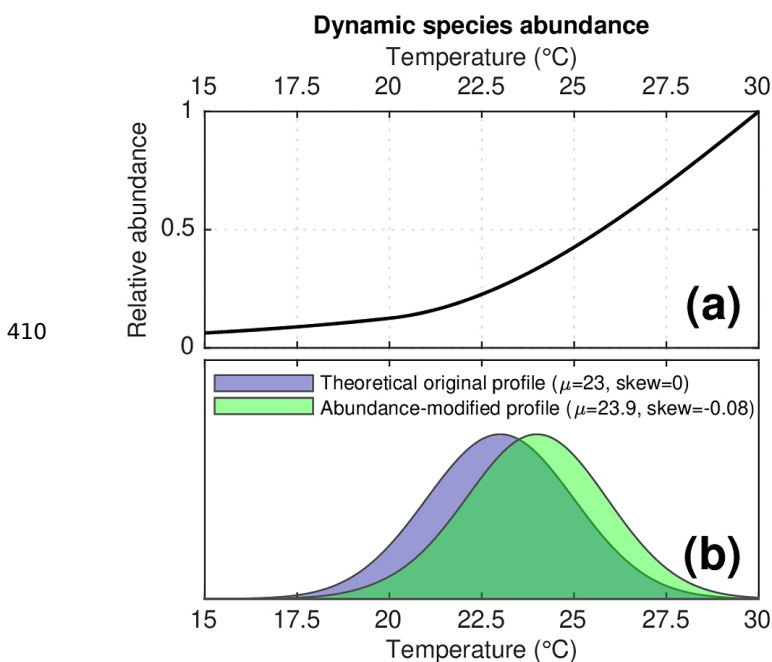

**Figure 2.** Panel a: The dynamic species abundance function applied to some of the simulations in this study. Panel b: An theoretical example of how the dynamic species abundance would bias recording of SST by individual foraminifera. In blue, a normally distributed theoretical SST profile. In green, the signal that would be recorded by a species affected by the dynamic species abundance function.

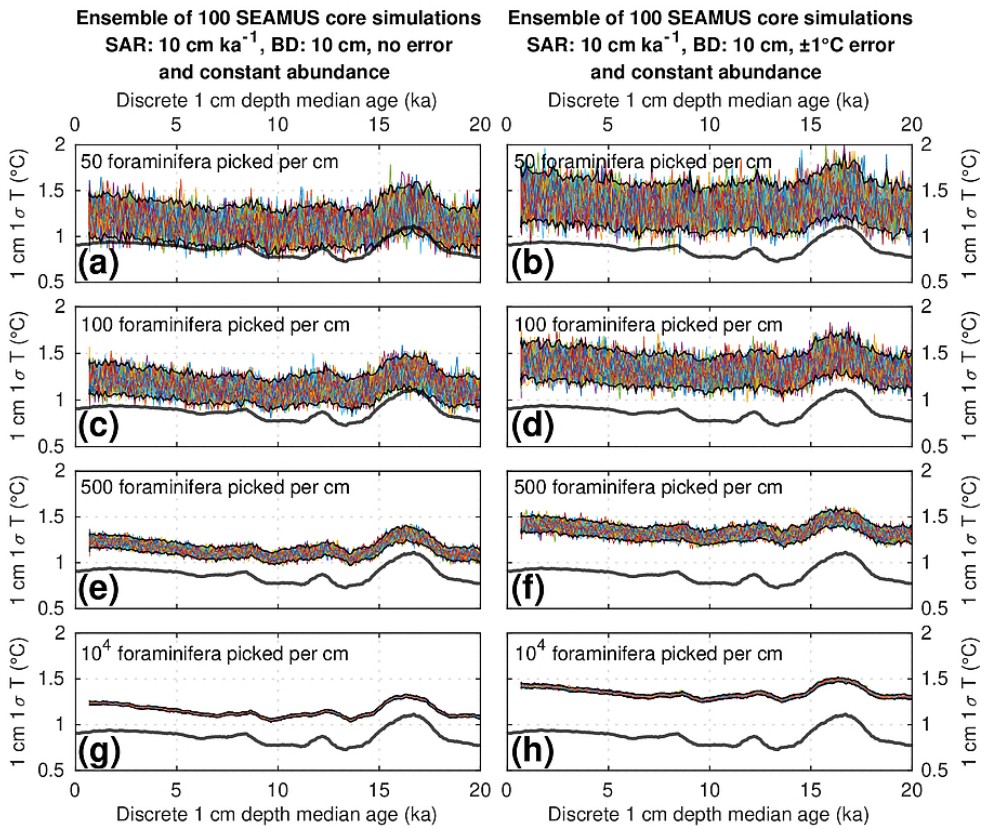


**Figure 3.** Simulated downcore, discrete 1 cm depth 1σ SST values of simulated single foraminifera from various 10 cm ka[-1] SAR scenarios with 10 cm BD, each with 100 ensembles of SEAMUS runs. In each panel, each ensemble is shown using a coloured line. The solid black lines represent the 95% interval of the ensemble runs at each discrete 1 cm depth. Also shown for reference as a thick grey

line is the 1000 year (12000 month) moving 1σ of the 1.5-7 year filtered monthly SST data (as also shown in Fig. 1c.) The left panels (a, c, e and g) show the output of scenarios with 50, 100, 500 and 10[4] randomly picked foraminifera per discrete 1 cm depth, all with constant species abundance and no assumed analytical error. The right panels (b, d, f and h) show the output of scenarios with 50, 100, 500 and 10[4] randomly picked foraminifera per discrete 1 cm depth, all with constant species

abundance and an assumed analytical error of ±1°C in SST.

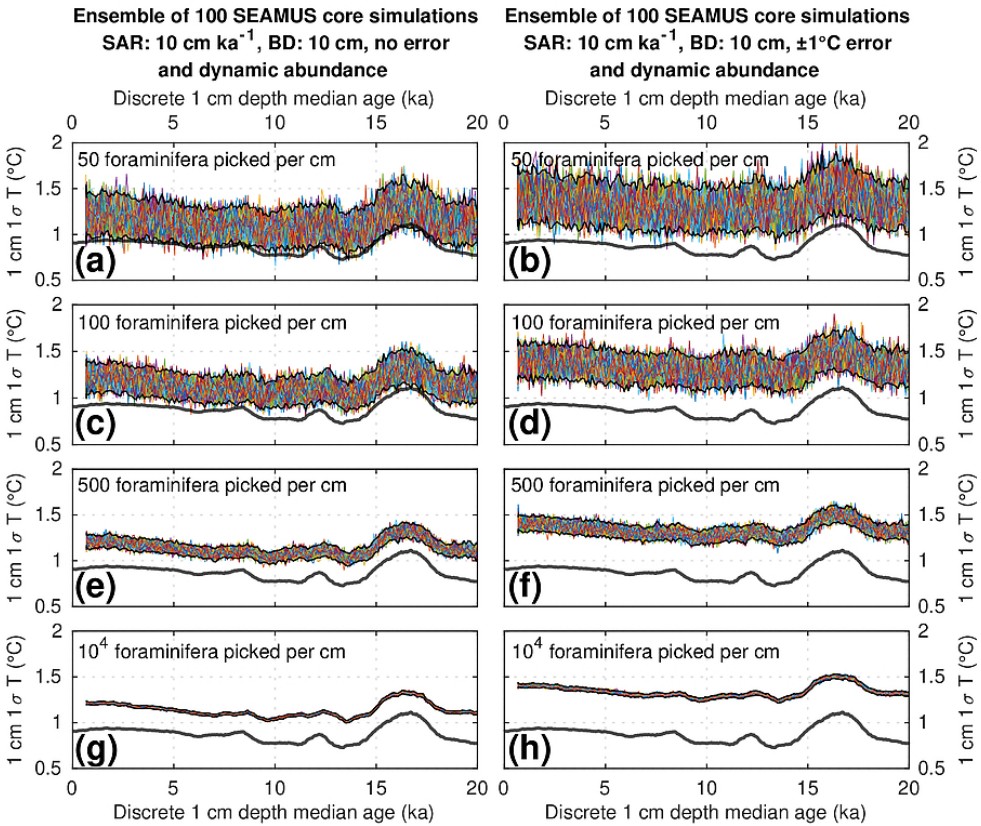

**Figure 4.** Simulated downcore, discrete 1 cm depth 1σ SST values of simulated single foraminifera from various 10 cm ka⁻¹ SAR scenarios with 10 cm BD, each with 100 ensembles of SEAMUS runs. In each panel, each ensemble is shown using a coloured line. The solid black lines represent the 95% interval of the ensemble runs at each discrete 1 cm depth. Also shown for reference as a thick grey line is the 1000 year (12000 month) moving 1σ of the 1.5-7 year filtered monthly SST data (as also shown in Fig. 1c. The left panels (a, c, e and g) show the output of scenarios with 50, 100, 500 and $10^4$ randomly picked foraminifera per discrete 1 cm depth, all with dynamic species abundance (following Fig. 2a) and no assumed analytical error. The right panels (b, d, f and h) show the output of scenarios with 50, 100, 500 and $10^4$ randomly picked foraminifera per discrete 1 cm depth, all with dynamic species abundance (following Fig. 2a) and an assumed analytical error of ±1°C in SST.





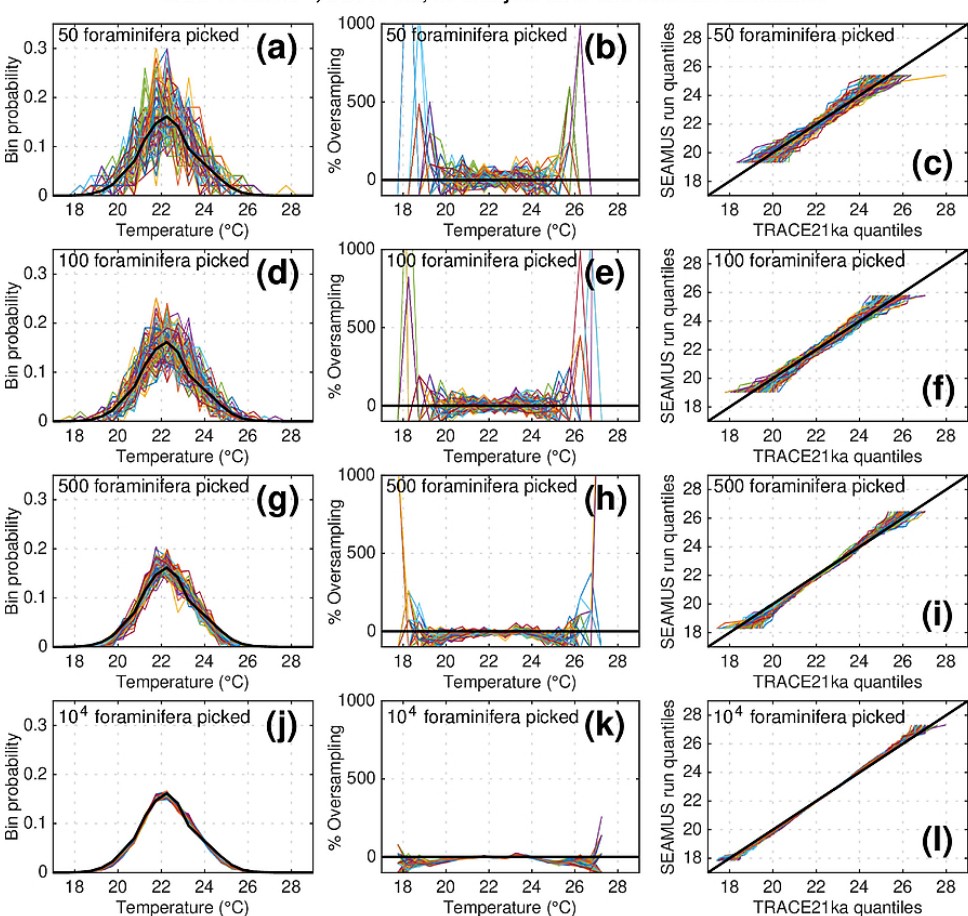

**Figure 5.** Simulated single foraminifera SST distributions from 100 ensembles of SEAMUS runs, with SAR of 10 cm ka$^{-1}$, BD of 10 cm, no analytical errror and constant abundance. In each ensemble, the single foraminifera SST distribution from a single discrete depth with a simulated median age of 17.5 ka is shown, and compared to the TRACE-21ka SST distribution for the 18 ka to 17 ka period. The left panels (a, d, g and j) show the 100 SEAMUS ensembles as coloured lines in the case of 50, 100, 500 and 10$^4$ randomly picked foraminifera, with the TRACE-21ka SST distribution is shown as a black line. The middle panels (b, e, h and k) show the the rate of over/undersampling for each of the 100 SEAMUS ensembles (coloured lines) relative to the TRACE-21ka SST distribution (black line) in the case of 50, 100, 500 and 10$^4$ randomly picked foraminifera. The right panels (c, f, i and l) show Q-Q plots of the 100 SEAMUS ensemble quantiles vs the TRACE-21ka quantiles as coloured lines in the case of 50, 100, 500 and 10$^4$ randomly picked foraminifera, with a perfect 1:1 correspondence to TRACE-21ka shown for reference as a black line.





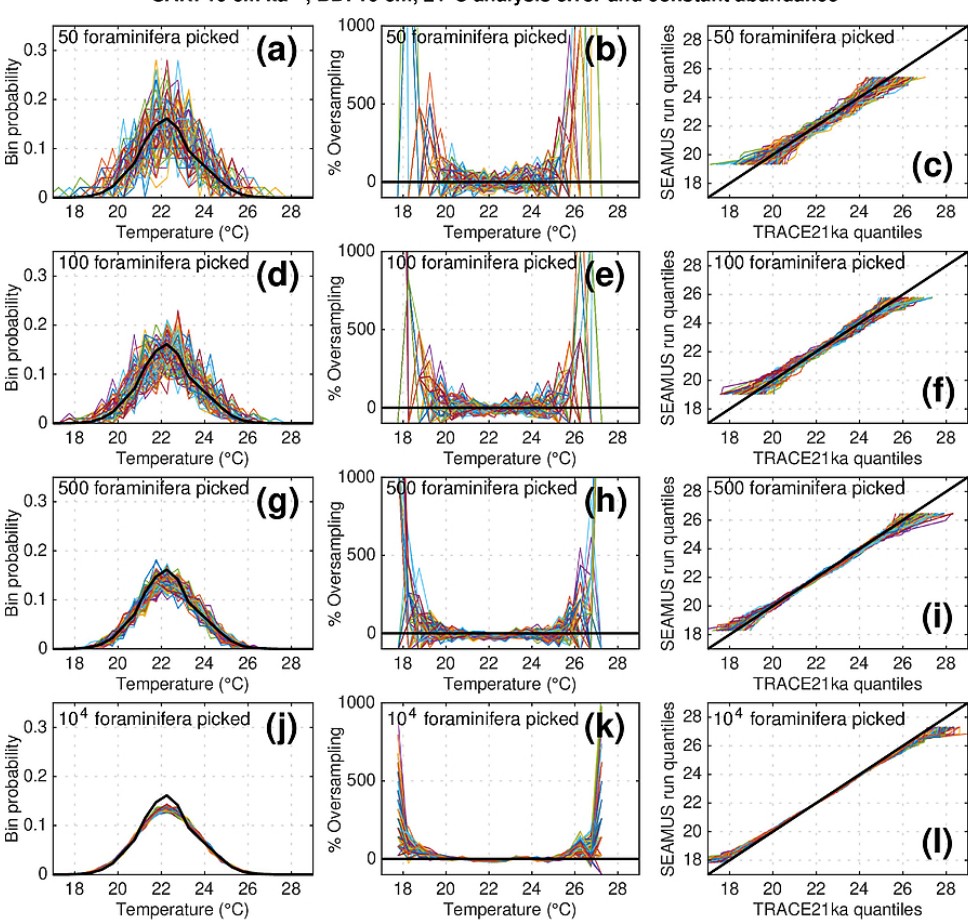


**Figure 6.** Simulated single foraminifera SST distributions from 100 ensembles of SEAMUS runs, with SAR of 10 cm ka$^{-1}$, BD of 10 cm, ±1 °C analytical errror and constant abundance. In each ensemble, the single foraminifera SST distribution from a single discrete depth with a simulated median age of 17.5 ka is shown, and compared to the TRACE-21ka SST distribution for the 18 ka to 17 ka period. The left panels (a, d, g and j) show the 100 SEAMUS ensembles as coloured lines in the case of 50, 100, 500 and 10$^4$ randomly picked foraminifera, with the TRACE-21ka SST distribution is shown as a black line. The middle panels (b, e, h and k) show the the rate of over/undersampling for each of the 100 SEAMUS ensembles (coloured lines) relative to the TRACE-21ka SST distribution (black line) in the case of 50, 100, 500 and 10$^4$ randomly picked foraminifera. The right panels (c, f, i and l) show Q-Q plots of the 100 SEAMUS ensemble quantiles vs the TRACE-21ka quantiles as coloured lines in the case of 50, 100, 500 and 10$^4$ randomly picked foraminifera, with a perfect 1:1 correspondence to TRACE-21ka shown for reference as a black line.







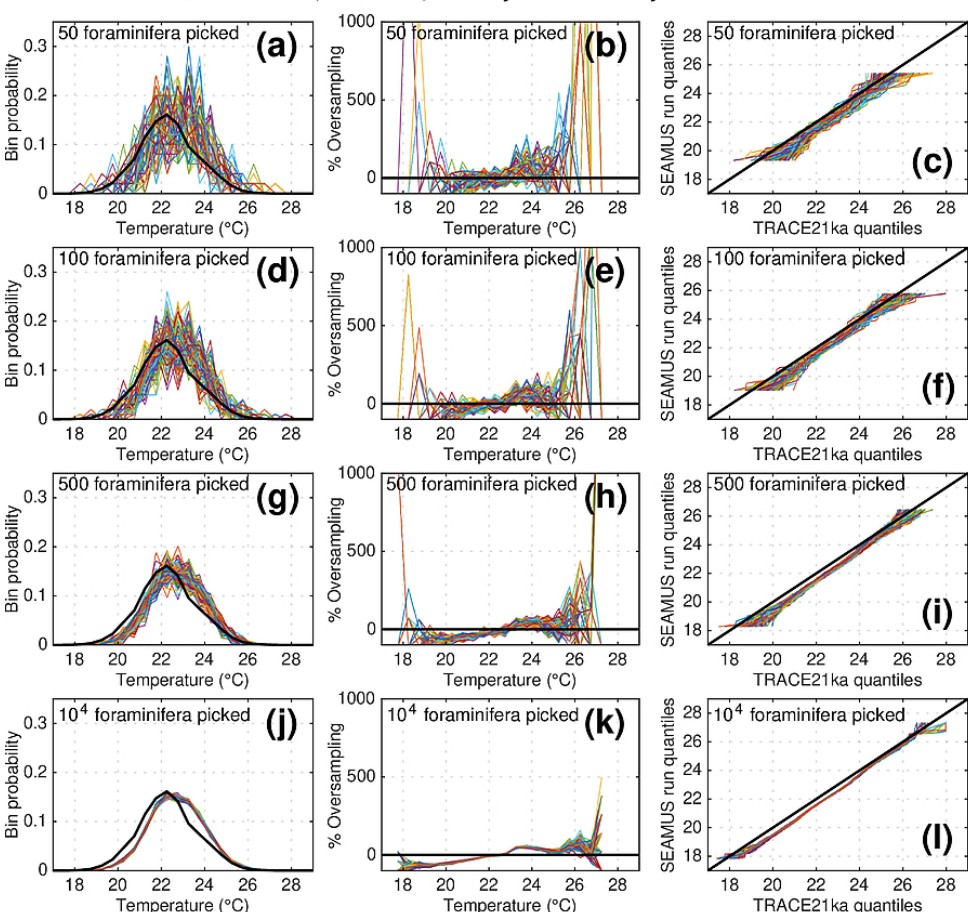

**Figure 7.** Simulated single foraminifera SST distributions from 100 ensembles of SEAMUS runs, with SAR of 10 cm ka$^{-1}$, BD of 10 cm, no analytical errror and dynamic abundance (following Fig. 2a). In each ensemble, the single foraminifera SST distribution from a single discrete depth with a simulated median age of 17.5 ka is shown, and compared to the TRACE-21ka SST distribution for the 18 ka to 17 ka period. The left panels (a, d, g and j) show the 100 SEAMUS ensembles as coloured lines in the case of 50, 100, 500 and 10$^4$ randomly picked foraminifera, with the TRACE-21ka SST distribution is shown as a black line. The middle panels (b, e, h and k) show the the rate of over/undersampling for each of the 100 SEAMUS ensembles (coloured lines) relative to the TRACE-21ka SST distribution (black line) in the case of 50, 100, 500 and 10$^4$ randomly picked foraminifera. The right panels (c, f, i and l) show Q-Q plots of the 100 SEAMUS ensemble quantiles vs the TRACE-21ka quantiles as coloured lines in the case of 50, 100, 500 and 10$^4$ randomly picked foraminifera, with a perfect 1:1 correspondence to TRACE-21ka shown for reference as a black line.



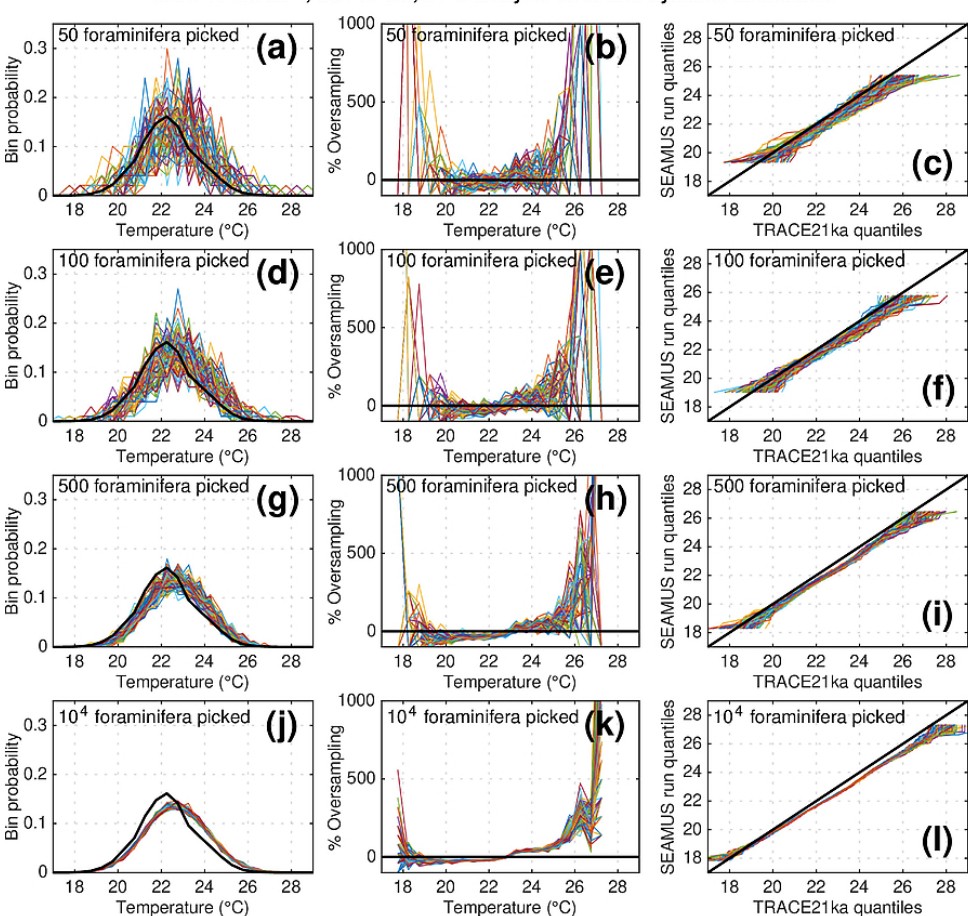

**Figure 8.** Simulated single foraminifera SST distributions from 100 ensembles of SEAMUS runs, with SAR of 10 cm ka$^{-1}$, BD of 10 cm, ±1 °C analytical errror and dynamic abundance (following Fig. 2a). In each ensemble, the single foraminifera SST distribution from a single discrete depth with a simulated median age of 17.5 ka is shown, and compared to the TRACE-21ka SST distribution for the 18 ka to 17 ka period. The left panels (a, d, g and j) show the 100 SEAMUS ensembles as coloured lines in the case of 50, 100, 500 and 10$^4$ randomly picked foraminifera, with the TRACE-21ka SST distribution is shown as a black line. The middle panels (b, e, h and k) show the the rate of over/undersampling for each of the 100 SEAMUS ensembles (coloured lines) relative to the TRACE-21ka SST distribution (black line) in the case of 50, 100, 500 and 10$^4$ randomly picked foraminifera. The right panels (c, f, i and l) show Q-Q plots of the 100 SEAMUS ensemble quantiles vs the TRACE-21ka quantiles as coloured lines in the case of 50, 100, 500 and 10$^4$ randomly picked foraminifera, with a perfect 1:1 correspondence to TRACE-21ka shown for reference as a black line.

