# Peer review of "Testing the effect of bioturbation and species abundance upon discrete-depth individual foraminifera analysis."

_Biogeosciences, 2021_

## Author Comment (AC1)

Thank you to the referee for providing helpful comments which will serve to improve the reader's experience of the manuscript. In the spirit of the EGU journal discussion forum format, we strive here to provide a response well before the discussion period has elapsed. We discuss the comments of the referee below, whereby the referee's comments are indented and in blue.

> In this manuscript, Lougheed and Metcalfe use transient model outputs (Trace21k output as input for SEAMUS) to assess whether discrete-depth Individual Foraminifera Analysis (IFA) can faithfully reflect temperature distribution. Within the idealized model environment, the authors are able to simulate pre-depositional, post-depositional and post-retrieval processes that may affect the temperature distribution recorded by foraminiferal tests becoming a part of the sediment, including sea surface temperature (SST), foraminiferal abundance in response to SST, sediment accumulation rate, bioturbation, number of foraminifera picked (sample size), and machine error. They assess the sensitivity of IFA-derived SST distribution by varying the aforementioned parameters in the model environment. The output is of course best-case scenario – as the reality is a lot more chaotic and noisy. Despite this, the idealized simulations show that the IFA-derived SST distribution show extremely low reproducibility with the typical sample size adopted by users of IFA (50-100 picked specimens). The reproducibility is especially poor near the edge of the distribution – which is the region of interest to paleoceanographers. Another important finding is that varying species abundance in response to climate change may also bias IFA-derived reconstructions, and this bias cannot be avoided if one were to pick 10000 specimens for the IFA-based reconstruction.

This is an accurate description of the work. We are happy that you were able to follow everything!

> IFA has become increasingly popular as a tool for reconstructing past climate variability, thus the scientific questions explored by the authors are timely and of broad appeal. The manuscript is clear, generally well-written and accessible even to readers who have no strong background in numerical modelling. I expect the paper to be of great interest to users of IFA and proxy system modelling, and to a lesser degree also to those who study foraminiferal ecology. The scope of the study also fits the remit of the journal. I find the conclusions convincing, but think that the paper may benefit from some clarification here and there, and more discussion on how to apply the knowledge derived from these idealized simulations to actual sediment records, or at least some concrete suggestions on what (not) to do when using IFA in reconstruction. After all, the community that will benefit the most from this paper is likely paleoceanographers who apply IFA (I certainly hope so), thus the more reason to make it as accessible as possible to this community. In this regard, the reader could use some elaboration on what would be the minimum requirement in terms of SAR, sample size / number of specimen picked. Any regions where IFA would work nicely or should be avoided?

We prefer not to make strong recommendations about minimum number of foraminifera to pick, minimum SAR, etc, because we prefer not to make 'one size fits all' and/or 'black box' recommendations that may not apply to the sediment conditions all sites, nor the particular goals of a particular study. What we do here is quantify the noise and/or biases that may exist for a number of SAR, abundance and sample size scenarios scenarios. Interested researchers can then simply consider their own study in the context of the possible noise/bias.

Alternatively, we would encourage researchers to follow our modelling approach for the conditions at the particular site(s) they are working at, and perhaps define themselves the level of noise they could expect and to put their own results into context that way.

For discussions about which locations foraminifera populations in the water domain may or may not continuously record SST dynamics indicative of, e.g. ENSO, we refer to Metcalfe et al. (2020).

> Some comments on how realistic the idealized model output is, considering that one of the largest sources of uncertainty in IFA-derived SST distribution, i.e. the vertical migration of planktic foraminifera, is not considered in the simulation?

The model setup is intentionally idealised, and this is also the advantage of models, in that we can run a scenario with constant sedimentation rate, constant bioturbation depth, constant abundance, forams that constantly live at the sea surface, etc. This enables us to to test the method at the most fundamental level. It follows that if the method has issues in idealised conditions, that it will not perform better in real world conditions. If we were to run all model parameters as dynamic (not constant), it would not be possible to independently quantify the contribution of the various parameters to the noise/bias.

Unfortunately the TRACE21ka sea temperature was only available for the surface layer, so we could not investigate vertical water migration issues of the forams. However, in this paper we specifically seek to investigate and quantify the bias/noise caused by abundance changes and sedimentological issues (bioturbation), as stated in the title.

> I think it might also help maximize the impact of the work if the authors can make the outcome accessible in the form of a web GUI for users who are fluent in programming language – but this is more of a would-be-nice-to-have kind of suggestion.

Indeed, an online interactive GUI would be very interesting but would require very significant work to realise. The SEAMUS function is fully documented, and a walk through example is also bundled with SEAMUS. We would also be happy to help anybody get the script up and running. We would also point out that SEAMUS is fully Octave compatible, so a Matlab license would not be necessary.

> That said, I am happy to recommend publication once these concerns have been addressed by the authors. Altogether this should amount to minor to moderate revision. Below I outline a few specific comments / suggestions that I hope the authors will find helpful in revising their manuscript.

Thank you for your helpful comments, they will certainly help to improve the manuscript.

> Specific comments

> Line 29-32: Most studies are based on 50-100 specimens, so I'd add a sentence saying under what conditions can this sample size yield meaningful reconstruction. The results clearly indicate that one would be safe if 10000 specimens are picked, but alas this is not something that is realistic. Even 500 specimens are not always possible if one tries very hard.

We are aware that picking 10000 specimens is unrealistic, and we should indeed make this clearer in the manuscript, thank you for pointing this out. The reason we have a scenario with 10000 specimens per sample is to take advantage of the elegance of the computer simulation environment to include a reference scenario that is virtually free of "sample size noise".

Line 155-156: "… differs in model execution…" please elaborate more in what way it is different that makes it suitable for use in this IFA experiment.

The stochastic model explicitly simulates very many single elements (e.g. forams) in the sediment, whereas other bioturbation models are probabilistic, i.e. they predict the distribution of values for an entire population. There are two main advantages to using SEAMUS: (1) sample size noise and bioturbation noise are captured directly (relevant for this study). (2) it is possible to input all input variables as temporally dynamic if so desired. As far as I know, existing probabilistic bioturbation simulations require either one of sedimentation rate or abundance to be kept temporally constant.

The main disadvantage of SEAMUS is that explicitly simulating very many single elements requires significantly more computation time and memory, especially when multiple ensembles need to be run to fully quantify the noise of all the processes.

Line 163-165: This is a bit confusing. The model is run at monthly time-step, so the foraminifera in the model do not record daily temperature but only the monthly mean? Also, it would be helpful for the reader to follow the manuscript if the authors could provide more information on how the recording process is simulated in SEAMUS. Is the temperature value recorded by foraminiferal test an average of several weeks of daily temperature?

As is the case with palaeoclimate model runs, the TRACE21ka SST data is available in monthly resolution. SEAMUS can be run using any temporal resolution desired, and here we have run the SEAMUS model at monthly resolution to match TRACE21ka's monthly resolution. Each month SEAMUS simulates $n$ new forams, and these forams are assigned the SST value of that month in TRACE21ka, which is indeed the average temperature of the month.

Line 168: 10000 foraminifera per cm of sediment (at a single site presumably) sounds a lot. Is this value based on some ecological studies? If yes please add the references here. I also wonder if this value affects the model output? Say, for example, if one were to assume that only 1000 foraminifera are produced per cm of sediment, how would the smaller number of foraminifera affects the simulation of SST distribution.

Thank you, we will make clearer that 10000 forams per cm is simply taking advantage of the computer modelling environment to simulate many forams so that we can subsequently have a 10000 forams per cm picking scenario, which represents a "sample size noise-free" reference scenario.

We will also make clear that 10000 forams per cm in the sediment simulation does not affect the smaller sample size picking scenarios. In other words, simulating 10000 forams per cm and subsequently picking 50 forams per cm results in the same process outcome as simulating only 50 forams per cm in the sediment and picking all 50 simulated forams per cm.

Line 255-260: Why the criterion of r2>0.6 on top of p<0.05? Any reason why both criteria are needed for this study? I note that p<0.05 is a more commonly adopted criterion in paleoclimatology when assessing correlation between time series. How does the result change when only the p<0.05 criterion is applied?

Common practice is to also consider $p$ when investigating the Pearson correlation coefficient r, as far as we know. Considering p in isolation would lead to false confidence in the presence of a correlation in a case where, e.g., $p \leq 0.05$, but $r^2 = 0.1$.

Pearson's $r$ correlation coefficient indicates the strength of the correlation and its associated $p$ value indicates whether or not the $r$ correlation coefficient e is significant (discernable from noise). We compare the simulated downcore SST variance to the climate model SST variance to see if there is a strong and significant correlation. We use an $r^2$ threshold of 0.6, which would indicate that 60% (i.e. more than half) of the variation is common between the two variables. A t-test is subsequently used to calculate the $p$ value associated with the the $r$ coefficient.

We will make our motivation clearer in the final version of the manuscript, specifically by referring to both strength and significance of correlation, which we neglected to do in the text.

> Line 310-314: As this is one of the main results, please provide more detail on the calculation of over-sampling (e.g. what does it mean with >500% oversampling).

Thanks for pointing this out, we will explain this in the text.

> Line 356-357: I applaud the authors for being candid but this could be rephrased to sound a bit more positive. Something along the line of "results are associated with large uncertainties due to unconstrained model parameters". This will also set up nicely the next sentence about future work to quantify the errors associated with IFA-based reconstruction.

line 356-357: *"Consequently, our model results may either over- or understate challenges relevant to IFA."* Yes, we are essentially saying here that we are not fully familiar with the particular bioturbation depth and sedimentation rate values at all of the study sites of all other researchers, hence they would need to investigate these values at their site themselves, and how it may affect their findings. We disagree that our model parameters are unconstrained, they are 100% constrained (we know exactly what inputs went into the model).

> Section 4.0 (alternatively, add a new sub-section before section 4.0): See my general comments above. After going through the rather negative results based on the idealized simulations, one is left wondering what does this mean for real-world reconstruction. Can we indeed apply what we learn from this idealized simulation to actual records? Is it too early to tell, since the parameters used in the simulation are unconstrained? I think the reader, especially users of IFA, would like to have more details in this regard, as well as some concrete suggestions on what to do/ not to do when interpreting IFA-based reconstruction beyond the rather general suggestions already offered by the authors.

How can researchers overcome bioturbation's effect upon IFA? We have thought long and hard about how one could go about correcting particular studies for the inherent noise/bias associated with the aforementioned processes, but have yet to come up with a solution. Furthermore, we intentionally want to avoid cast iron guidance to researchers because we do not claim to have a silver bullet nor know all the answers. It is possible that bioturbation/sedimentological issues may be insurmountable in some cases. However, understanding and quantifying the possible sources of uncertainty in the sediment (as our study seeks to do) is an important first step in putting results in context and will help us to move beyond treating the sediment archive as a sequence of discrete age intervals (such as, e.g., tree rings are). This will help researchers to know with more confidence when they may or may not encounter significant results. Hence, we propose the following, as listed in the conclusion:

(1) Researchers should attempt quantify temporal trends in sediment accumulation rate, bioturbation

depth and species abundance at their site. Are these processes static, and how could they be affected by palaeoclimate itself?

(2) Run a forward model bioturbation study such as that detailed in this study, but using the aforementioned parameters from the site of interest, to determine the overall level of noise and/or SST bias at the site in question.

(3) Consider whether the total uncertainty/bias estimated from steps (1) & (2) has consequences for the interpretation of the data.

(4) Carry out replication studies (e.g. sample again from a second core from nearby).

---

## Author Comment (AC2)

We would like to thank the referee for their review, which will serve to improve the manuscript. We respond to the referee below, with the referees comments indented and in blue, and our response in black.

> The paper is very well written and well presented. The content should be of large interest for the relevant scientific community. Their results may be seen as provoking if you work on IFA, however they clearly express strengths and limitations of the work they present, and stress the need for evaluating the conditions (SAR, BD, foraminiferal abundance) at the given sites investigated. Overall, I find it to be a very good paper, more or less ready for publication. I only have a few minor comments.

Thank you for your kind words, we are glad that you note that we do our best to underline both the strengths and limitations of our experimental setup.

> Consider to reduce the number of abbreviations used to ease the reading.

We will try to do so, thank you for pointing it out.

> Add some information on the choice of site, and the real-life conditions you could expect at this site. The selected site is from a site where it would be great to have reconstructions, but where there is little foraminiferal material available for investigation.

Yes, we have assumed a "best case scenario" where it would be possible to retrieve a foram-rich sediment core from a location that is climatologically most interesting for ENSO, i.e. where the palaeo-ENSO signal could be expected to be very strong. Essentially, we want to test IFA under a hypothetical best case conditions. Any difficulties that exist in the "best-case" location would obviously be even more challenging in less optimal locations. We will make this clearer in the final version of the manuscript.

> SST is defined twice in the abstract (L13 and L16).

Oops, thanks!

> L63: not intuitively clear sentence. Suspect you refer to absolute abundance? and/or relative? Both will change through time.

Yes, here in the introduction we refer to abundance in general… which indeed changes through time no matter how it is defined.

> L217: if 5 cm/ka is more realistic for the area, why put the main focus son the 10cm/ka scenario? I read this as a theoretical paper with an idealized approach, forcing the foraminiferal model with modelled SSTs that again are used to verify the modelled IFA response. I see the argument for choosing this location, being a sensitive area in the modelled SST fields, does it really matter for your result what the realistic sed rate is in the area?

We use ensemble runs to model 5, 10 and 40 cm/ka. 10 cm/ka is included in the main text, with 5 and 40 cm/ka in the supplement. Indeed, any of the three could be argued to be included in the main text, and in various drafts we had others in the main text. So we settled on the intermediate scenario.

> L230: calculated as a by - delete as a

> L280: delete is

Thank you for noting these typos.

Could you consider different species and how responses may vary between species, or is the result species independent and more a general representation of the effects of SAR/BD on the IFA?

In Figures 7 and 8 it is shown that the modelled species with a preference for warmer temperatures shows a bias towards warmer temperatures in the IFA temperature reconstruction. In the case of a species with a preference for colder temperatures, the bias in the IFA temperature reconstruction would therefore be towards colder temperatures. We can mention this in the text for clarity, thank you for your comment.

I see that it would be out of the scope for this manuscript, but I have one suggestion for a future study that I would love to see and that I think would strengthen the message in the end. It would be very nice to see how this tool and analysis would compare to a "real" IFA study, given that the conclusion put strong constraints to the IFA approach. E.g., a study where the modelled data was from a location and a time interval where IFA have been/could be done, and from where the temperature is well known (for instance instrumental constraints on temperature, lead or marker horizons constrained ages and hence sedimentation/accumulation rates, stained foraminifera for information on living dept, if not BD). I think such a study would provide a more approachable message for many working on these issues since this paper presents a quite technical and theoretical approach to IFA analysis and results. And given that the message would be the same, further emphasis the potential issues that exists with respect to IFA approaches.

Yes, these would be interesting future studies. We note, however, that we cannot model the effect of bioturbation using the instrumental record because the instrumental record is not long enough to represent the interval of time incorporated by bioturbation.

For a study using the instrumental record to study the response of single foraminifera in the water domain and how they might record climate events such as ENSO and at which locations, we refer you to Metcalfe et al. (2020) [https://doi.org/10.5194/cp-16-885-2020].

Thanks again for taking the time to review our work!

---

## Author Response (AR1)

January 20, 2022.
Uppsala, Sweden.

Dear Hiroshi Kitazato,

Thank you once again for considering our manuscript for publication. The reviewers have raised some interesting points which have helped us better explain some points in the revised version of the manuscript.

As you requested, we have also added some more information about the importance of sample size to the introduction, so that the reader understands why we look at different sample size scenarios.

For your convenience, we also attach a "track changes" version of the manuscript.

Kind regards,

Bryan Lougheed